# Some Applications of Eigenvalue Problems for Tensor and Tensor–Block Matrices for Mathematical Modeling of Micropolar Thin Bodies

**Mikhail Nikabadze [1,2,\*] and Armine Ulukhanyan [2]**

[1]   Faculty of Mechanics and Mathematics, Lomonosov Moscow State University, 119991 Moscow, Russia
[2]   Department of Computational Mathematics and Mathematical Physics, Bauman Moscow State Technical
     University, 105005 Moscow, Russia; armine_msu@mail.ru
[\*]   Correspondence: nikabadze@mail.ru

**Abstract:** The statement of the eigenvalue problem for a tensor–block matrix (TBM) of any order and of any even rank is formulated, and also some of its special cases are considered. In particular, using the canonical presentation of the TBM of the tensor of elastic modules of the micropolar theory, in the canonical form the specific deformation energy and the constitutive relations are written. With the help of the introduced TBM operator, the equations of motion of a micropolar arbitrarily anisotropic medium are written, and also the boundary conditions are written down by means of the introduced TBM operator of the stress and the couple stress vectors. The formulations of initial-boundary value problems in these terms for an arbitrary anisotropic medium are given. The questions on the decomposition of initial-boundary value problems of elasticity and thin body theory for some anisotropic media are considered. In particular, the initial-boundary problems of the micropolar (classical) theory of elasticity are presented with the help of the introduced TBM operators (tensors–operators). In the case of an isotropic micropolar elastic medium (isotropic and transversely isotropic classical media), the TBM operator (tensors–operators) of cofactors to TBM operators (tensors–tensors) of the initial-boundary value problems are constructed that allow decomposing initial-boundary value problems. We also find the determinant and the tensor of cofactors to the sum of six tensors used for decomposition of initial-boundary value problems. From three-dimensional decomposed initial-boundary value problems, the corresponding decomposed initial-boundary value problems for the theories of thin bodies are obtained.

**Keywords:** tensor–operator of equations; stress tensor–operator; tensor–operator of stress and couple stress; tensor–block matrix operator; canonical presentation of tensor–block matrix; eigenoperator

**MSC:** 15A72; 47A75; 74J05; 74J10; 74B05; 74E10; 74E15; 53A45

## 1. Introduction

For isotropic materials, eigenmodules (eigenvalues) and eigenstates (eigentensors) are known since monograph [1]. For anisotropic ones, the said notions were introduced by Kelvin in the middle of the 19th century (other terms were used). However, the investigation in this area was continued only about 40 years ago (see, e.g., [2–15] and the bibliography of [10]). In [10], Ostrosablin has studied the inner structure rather well, classified anisotropic classically linearly elastic materials, and studied many other important problems (see also [16]). The notion of elastic eigenstates is applied in the plasticity theory (see [17]) and the flow theory (see [18]).

Representations of general solutions of the Lamé's equations were made by many scientists (see, for example, [19–23]), and representations of general solutions of equations in displacements and rotations

of the micropolar theory of elasticity can be found, for example, in [22,24–26]. Note that in this case the equations of the classical and micropolar theory of elasticity are decomposed, but for decomposing the equations, as well as static boundary conditions, the algebraic method turned out to be more efficient. If this method is used, it is advisable to present the equations and static boundary conditions in tensors–operators (tensor–block operators) in the case of classical (micropolar) medium, and then find the tensors–operators (tensor–block operators) of algebraic cofactors for these operators. Of course, the algebraic method can be used for decomposing the static boundary conditions only for bodies with piecewise plane boundaries. These questions are described in some detail in [27,28], and in this work, special attention is paid to the canonical representations of equations and boundary conditions.

Note that in [28] some questions from monograph [27], which was published in Russian in the Mechanics and Mathematics faculty of Lomonosov Moscow State University, were presented in English with some clarifications and changes. In particular, the authors presented some questions of tensor calculus; constructed new versions of theories of single-layer and multi-layer elastic thin bodies via the developed method of orthogonal polynomials and also obtained the corresponding decomposed equations of quasistatic problems of classical (micropolar) theory of prismatic bodies with constant thickness in displacements (in displacements and rotations) from the decomposed equations of classical (micropolar) theory of elasticity. The above results and eigenvalue problems for tensor and tensor–block matrices (see [29]) are used in this work for mathematical modeling of the micropolar thin bodies.

## 2. Statement of Eigenvalue Problem of a Tensor–Block Matrix of Any Even Rank

Find all tensor columns $\mathbb{U}$ which satisfy equation

$$\mathbb{M} \overset{p}{\otimes} \mathbb{U} = \lambda \mathbb{U},$$

where $\lambda$ is scalar, and

$$
\mathbb{U} = \begin{pmatrix} \mathbf{U}_1 \\ \mathbf{U}_2 \\ . \\ . \\ . \\ \mathbf{U}_m \end{pmatrix}, \quad
\mathbb{M} = \begin{pmatrix}
\mathbf{A}_{11} & \mathbf{A}_{12} & \mathbf{A}_{13} & \cdots & \mathbf{A}_{1m} \\
\mathbf{A}_{21} & \mathbf{A}_{22} & \mathbf{A}_{23} & \cdots & \mathbf{A}_{2m} \\
\mathbf{A}_{31} & \mathbf{A}_{32} & \mathbf{A}_{33} & \cdots & \mathbf{A}_{3m} \\
\cdots & \cdots & \cdots & \cdots & \cdots \\
\mathbf{A}_{m1} & \mathbf{A}_{m2} & \mathbf{A}_{m3} & \cdots & \mathbf{A}_{mm}
\end{pmatrix}, \tag{1}
$$

$$\mathbb{M} = \mathbb{M}_{i_1 i_2 \cdots i_p}^{j_1 j_2 \cdots j_p} \mathbf{R}^{i_1 \cdots i_p} \mathbf{R}_{j_1 \cdots j_p} = \mathbb{M}_i^j \mathbf{R}^i \mathbf{R}_j, \quad \mathbf{R}_{i_1 \cdots i_p} = \mathbf{r}_1 \otimes \cdots \otimes \mathbf{r}_p,$$

$$\mathbf{R}_i \overset{p}{\otimes} \mathbf{R}^j = g_i^j, \quad \mathbf{A}_{kl} = A_{kl, j_1 j_2 \cdots j_p}^{i_1 i_2 \cdots i_p} \mathbf{R}_{i_1 i_2 \cdots i_p} \mathbf{R}^{j_1 j_2 \cdots j_p} = A_{kl, \cdot j}^{\ i} \mathbf{R}_i \mathbf{R}^j,$$

$$\mathbf{U}_k = \mathbf{U}_{k, i_1 i_2 \cdots i_p} \mathbf{R}^{i_1 \cdots i_p}, \quad k, l = \overline{1, m}, \quad i_1, i_2, \ldots, i_p, \quad j_1, j_2, \ldots, j_p = \overline{1, n},$$

$$i, j = \overline{1, N}, \quad N = n^p.$$

Note that this problem is solved for the tensor of any even rank and the tensor–block matrix of any even rank consisting of four tensors, as well as for the tensor and the tensor–block matrix of the fourth rank, and published in [29]. Therefore, here we will not dwell on the presentation of this problem with the aim of shortening the letter, but, if necessary, we will refer to the work mentioned in the previous sentence. We also note that, solving the eigenvalue problem for a tensor–block matrix of any even rank consisting of four tensors, there is no difficulty in solving the analogous problem for the tensor–block matrix $\mathbb{M}$ (see (1)). Thus, we assume that the eigenvalue problem for a tensor–block matrix of any order and of any even rank is solved and we will consider some its applications below.

## 3. Equations of Motion Relative to the Displacement and Rotation Vectors for an Elastic Material without a Center of Symmetry

The constitutive relations (CR) given in [25,30,31] for a linearly elastic inhomogeneous anisotropic material without a center of symmetry for small displacements and rotations and isothermal processes can be written as

$$\underset{\sim}{\mathbf{P}} = \underset{\approx}{\mathbf{A}} \overset{2}{\otimes} \boldsymbol{\gamma} + \underset{\approx}{\mathbf{B}} \overset{2}{\otimes} \boldsymbol{\varkappa}, \quad \boldsymbol{\mu} = \underset{\approx}{\mathbf{C}} \overset{2}{\otimes} \boldsymbol{\gamma} + \underset{\approx}{\mathbf{D}} \overset{2}{\otimes} \boldsymbol{\varkappa} \quad (\boldsymbol{\gamma} = \nabla \mathbf{u} - \underset{\sim}{\mathbf{C}} \cdot \boldsymbol{\varphi}, \ \boldsymbol{\varkappa} = \nabla \boldsymbol{\varphi}), \tag{2}$$

where $\underset{\sim}{\mathbf{P}}$ and $\boldsymbol{\mu}$ are the stress and couple-stress tensors, $\boldsymbol{\gamma}$ and $\boldsymbol{\varkappa}$ are the tensors of deformation and bending-torsion, $\mathbf{u}$ and $\boldsymbol{\varphi}$ are the displacement and rotation vectors, $\underset{\approx}{\mathbf{A}}$, $\underset{\approx}{\mathbf{C}} = \underset{\approx}{\mathbf{B}}^T$ and $\underset{\approx}{\mathbf{D}}$ are the material tensors of the fourth rank, $\underset{\sim}{\mathbf{C}}$ is the discriminant tensor of third rank, $\overset{2}{\otimes}$ is the inner two-product [27,29,32–35], the superscript $T$ in the upper right corner of the quantities denotes transposition.

Introducing the tensor columns of the deformation and bending-torsion tensors and stress and couple-stress tensors, as well as the fourth rank tensor–block matrix (TBM) of the elastic modulus tensors

$$\underset{\approx}{\mathbb{X}} = \begin{pmatrix} \boldsymbol{\gamma} \\ \boldsymbol{\varkappa} \end{pmatrix} \ \left( \underset{\approx}{\mathbb{X}}^T = ( \ \boldsymbol{\gamma}, \boldsymbol{\varkappa} \ ) \right), \quad \underset{\approx}{\mathbb{Y}} = \begin{pmatrix} \underset{\sim}{\mathbf{P}} \\ \boldsymbol{\mu} \end{pmatrix} \ \left( \underset{\approx}{\mathbb{Y}}^T = ( \ \underset{\sim}{\mathbf{P}}, \boldsymbol{\mu} \ ) \right),$$

$$\underset{\approx}{\mathbb{M}} = \begin{pmatrix} \underset{\approx}{\mathbf{A}} & \underset{\approx}{\mathbf{B}} \\ \underset{\approx}{\mathbf{C}} & \underset{\approx}{\mathbf{D}} \end{pmatrix} \ \left( \underset{\approx}{\mathbb{M}}^T = \underset{\approx}{\mathbb{M}} \right), \tag{3}$$

the specific strain energy and the CR can be written in the form

$$2\Phi(\boldsymbol{\gamma}, \boldsymbol{\varkappa}) = \boldsymbol{\gamma} \overset{2}{\otimes} \underset{\approx}{\mathbf{A}} \overset{2}{\otimes} \boldsymbol{\gamma} + 2\boldsymbol{\gamma} \overset{2}{\otimes} \underset{\approx}{\mathbf{B}} \overset{2}{\otimes} \boldsymbol{\varkappa} + \boldsymbol{\varkappa} \overset{2}{\otimes} \underset{\approx}{\mathbf{D}} \overset{2}{\otimes} \boldsymbol{\varkappa} = \mathbb{X}^T \overset{2}{\otimes} \underset{\approx}{\mathbb{M}} \overset{2}{\otimes} \mathbb{X}, \quad \mathbb{Y} = \underset{\approx}{\mathbb{M}} \overset{2}{\otimes} \mathbb{X}. \tag{4}$$

If the material has a center of symmetry in the sense of elastic properties, then $\underset{\approx}{\mathbf{B}} = \underset{\approx}{\mathbf{0}}$, where $\underset{\approx}{\mathbf{0}}$ is the zero tensor of the fourth rank and the tensor–block matrix of the elastic modulus tensors (3) will take the form of a tensor–block-diagonal matrix.

Substituting (2) in the equations of motion for small displacements and rotations

$$\nabla \cdot \underset{\sim}{\mathbf{P}} + \rho \mathbf{F} = \rho \partial_t^2 \mathbf{u}, \quad \nabla \cdot \boldsymbol{\mu} + \underset{\sim}{\mathbf{C}} \overset{2}{\otimes} \underset{\sim}{\mathbf{P}} + \rho \mathbf{m} = \underset{\sim}{\mathbf{J}} \partial_t^2 \boldsymbol{\varphi},$$

and introducing the 2nd rank tensor–block matrix operator of the equations of motion and the vector columns of the displacement and rotation vectors and vectors of volume forces and moments

$$\underset{\sim}{\mathbb{M}} = \begin{pmatrix} \underset{\sim}{\mathbf{A}} & \underset{\sim}{\mathbf{B}} \\ \underset{\sim}{\mathbf{C}} & \underset{\sim}{\mathbf{D}} \end{pmatrix}, \quad \mathbb{U} = \begin{pmatrix} \mathbf{u} \\ \boldsymbol{\varphi} \end{pmatrix}, \quad \mathbb{X} = \begin{pmatrix} \rho \mathbf{F} \\ \rho \mathbf{m} \end{pmatrix}, \tag{5}$$

we obtain the equations of motion in displacements and rotations in the form

$$\underset{\sim}{\mathbb{M}} \cdot \mathbb{U} + \mathbb{X} = 0, \tag{6}$$

where the differential tensors–operators $\underset{\sim}{\mathbf{A}}$, $\underset{\sim}{\mathbf{B}}$, $\underset{\sim}{\mathbf{C}}$ and $\underset{\sim}{\mathbf{D}}$ have expressions

$$\underset{\sim}{\mathbf{A}} = \underset{\sim}{\mathbf{A}}' - \underset{\sim}{\mathbf{E}} \rho \partial_t^2, \quad \underset{\sim}{\mathbf{A}}' = \mathbf{r}_j \mathbf{r}_l (A^{ijkl} \nabla_i + \nabla_i A^{ijkl}) \nabla_k,$$

$$\underset{\sim}{\mathbf{B}} = \mathbf{r}_j \mathbf{r}_l [(B^{ijkl} \nabla_i + \nabla_i B^{ijkl} - C^l_{.mn} A^{mnkj}) \nabla_k - C^l_{.mn} \nabla_i A^{mnij}],$$

$$\underset{\sim}{\mathbf{C}} = \mathbf{r}_j \mathbf{r}_l (B^{klij} \nabla_i + \nabla_i B^{klij} + C^j_{.mn} A^{mnkl}) \nabla_k, \quad \underset{\sim}{\mathbf{D}} = \underset{\sim}{\mathbf{D}}' - \underset{\sim}{\mathbf{J}} \partial_t^2, \tag{7}$$

$$\underset{\sim}{\mathbf{D}}' = \mathbf{r}_j \mathbf{r}_l \{ [D^{ijkl} \nabla_i + \nabla_i D^{ijkl} + (g_s^j g_t^l - g_s^l g_t^j) C^s_{.mn} B^{mnkt}] \nabla_k$$
$$- C^l_{.pq} (A^{pqmn} C^{.j}_{mn.} + \nabla_i B^{pqij}) \}.$$

Here and below, $\underset{\sim}{\mathbf{E}}$ is the unit tensor of the second rank, $t$ is the time, and $\partial_t$ is the partial derivative operator with respect to time.

If we solve the eigenvalue problem for the tensor–block matrix (3), then we obtain its canonical presentation [29,34]

$$\underset{\approx}{\mathbb{M}} = \sum_{p=1}^{18} \lambda_p \mathbb{W}_p \otimes \mathbb{W}_p^T = \sum_{p=1}^{18} \lambda_p \begin{pmatrix} \mathbf{u}_p \otimes \mathbf{u}_p & \mathbf{u}_p \otimes \mathbf{v}_p \\ \mathbf{v}_p \otimes \mathbf{u}_p & \mathbf{v}_p \otimes \mathbf{v}_p \end{pmatrix}, \tag{8}$$

where $\lambda_p$, $p = \overline{1,18}$, are eigenvalues for positive-definite TBM from Equation (3). Moreover, $\lambda_1 \geq \lambda_2 \geq \lambda_3 \geq ... \geq \lambda_{18} > 0$, and $\mathbb{W}_p = (\mathbf{u}_p, \mathbf{v}_p)^T$, $p = \overline{1,18}$, is a complete orthonormal system of eigentensor columns satisfying the orthonormality conditions [29,34]

$$(\mathbb{W}_p, \mathbb{W}_q) = \mathbb{W}_p^T \overset{2}{\otimes} \mathbb{W}_q = \mathbf{u}_p \overset{2}{\otimes} \mathbf{u}_q + \mathbf{v}_p \overset{2}{\otimes} \mathbf{v}_q = \delta_{pq}, \quad p,q = \overline{1,18}. \tag{9}$$

Taking into account the expressions for TBM in Equation (3), from Equation (8) we get

$$\underset{\approx}{\mathbf{A}} = \sum_{p=1}^{18} \lambda_p \mathbf{u}_p \mathbf{u}_p, \quad \underset{\approx}{\mathbf{B}} = \underset{\approx}{\mathbf{C}}^T = \sum_{p=1}^{18} \lambda_p \mathbf{u}_p \mathbf{v}_p, \quad \underset{\approx}{\mathbf{D}} = \sum_{p=1}^{18} \lambda_p \mathbf{v}_p \mathbf{v}_p.$$

It is not difficult to prove that the inverse TBM $\underset{\approx}{\mathbb{M}}^{-1}$ to (8) has the form

$$\underset{\approx}{\mathbb{M}}^{-1} = \sum_{p=1}^{18} \lambda_p^{-1} \mathbb{W}_p \otimes \mathbb{W}_p^T = \sum_{p=1}^{18} \lambda_p^{-1} \begin{pmatrix} \mathbf{u}_p \otimes \mathbf{u}_p & \mathbf{u}_p \otimes \mathbf{v}_p \\ \mathbf{v}_p \otimes \mathbf{u}_p & \mathbf{v}_p \otimes \mathbf{v}_p \end{pmatrix}. \tag{10}$$

Generally speaking, for any integer $\alpha$ we have

$$\underset{\approx}{\mathbb{M}}^\alpha = \sum_{p=1}^{18} \lambda_p^\alpha \mathbb{W}_p \otimes \mathbb{W}_p^T = \sum_{p=1}^{18} \lambda_p^\alpha \begin{pmatrix} \mathbf{u}_p \otimes \mathbf{u}_p & \mathbf{u}_p \otimes \mathbf{v}_p \\ \mathbf{v}_p \otimes \mathbf{u}_p & \mathbf{v}_p \otimes \mathbf{v}_p \end{pmatrix}. \tag{11}$$

In particular, from (11) for $\alpha = 0$ we have

$$\underset{\approx}{\mathbb{E}} = \underset{\approx}{\mathbb{M}}^0 = \sum_{p=1}^{18} \mathbb{W}_p \otimes \mathbb{W}_p^T = \sum_{p=1}^{18} \begin{pmatrix} \mathbf{u}_p \otimes \mathbf{u}_p & \mathbf{u}_p \otimes \mathbf{v}_p \\ \mathbf{v}_p \otimes \mathbf{u}_p & \mathbf{v}_p \otimes \mathbf{v}_p \end{pmatrix}, \tag{12}$$

where $\underset{\approx}{\mathbb{E}}$ is the unit TBM of the fourth rank relative to the operations of the inner 2-product. Given (8), from (4) we get

$$2\Phi(\underset{\sim}{\boldsymbol{\gamma}}, \underset{\sim}{\boldsymbol{\varkappa}}) = \sum_{p=1}^{18} \lambda_p \mathbb{X}^T \overset{2}{\otimes} \mathbb{W}_p \mathbb{W}_p^T \overset{2}{\otimes} \mathbb{X}, \quad \mathbb{Y} = \sum_{p=1}^{18} \lambda_p \mathbb{W}_p \mathbb{W}_p^T \overset{2}{\otimes} \mathbb{X}. \tag{13}$$

Multiplying both sides of the second relation of (13) scalarly by $\mathbb{W}_\alpha$ and taking into account (9), the CR can be written in the form

$$(\mathbb{Y}, \mathbb{W}_\alpha) = \lambda_\alpha (\mathbb{X}, \mathbb{W}_\alpha) \; (\mathbb{Y}^T \overset{2}{\otimes} \mathbb{W}_\alpha = \lambda_\alpha \mathbb{X}^T \overset{2}{\otimes} \mathbb{W}_\alpha), \; \langle \alpha = \overline{1,18} \rangle. \tag{14}$$

Note that the formulas in Equation (14) give equivalent records of the CR. Introducing the notations

$$\mathbb{X}_\alpha = (\mathbb{X}, \mathbb{W}_\alpha) = \mathbb{X}^T \overset{2}{\otimes} \mathbb{W}_\alpha = \mathbb{W}_\alpha^T \overset{2}{\otimes} \mathbb{X} = \mathbf{u}_\alpha \overset{2}{\otimes} \underset{\sim}{\boldsymbol{\gamma}} + \mathbf{v}_\alpha \overset{2}{\otimes} \underset{\sim}{\boldsymbol{\varkappa}},$$

$$\mathbb{Y}_\alpha = (\mathbb{Y}, \mathbb{W}_\alpha) = \mathbb{Y}^T \overset{2}{\otimes} \mathbb{W}_\alpha = \mathbb{W}_\alpha^T \overset{2}{\otimes} \mathbb{Y} = \mathbf{u}_\alpha \overset{2}{\otimes} \underset{\sim}{\mathbf{P}} + \mathbf{v}_\alpha \overset{2}{\otimes} \underset{\sim}{\boldsymbol{\mu}}, \; \langle \alpha = \overline{1,18} \rangle, \tag{15}$$

the specific deformation energy (the first equality of (13)) and the CR (14) are represented in the form

$$2\Phi(\underset{\sim}{\boldsymbol{\gamma}}, \underset{\sim}{\boldsymbol{\varkappa}}) = \sum_{p=1}^{18} \lambda_p \mathbb{X}_p^2, \quad \mathbb{Y}_\alpha = \lambda_\alpha \mathbb{X}_\alpha, \quad \langle \alpha = \overline{1,18} \rangle.$$

Multiplying both sides of the first equality of (15) from the left tensorly by $\underline{\mathbb{W}}_\alpha$, and then if we sum the resulting ratio from 1 to 18, by (12) we get

$$\begin{aligned} \mathbb{X} &= \sum_{\alpha=1}^{18} \mathbb{X}_\alpha \underline{\mathbb{W}}_\alpha = \sum_{\alpha=1}^{18} (\underline{\mathbf{u}}_\alpha \overset{2}{\otimes} \underset{\sim}{\boldsymbol{\gamma}} + \underline{\mathbf{v}}_\alpha \overset{2}{\otimes} \underset{\sim}{\boldsymbol{\varkappa}}) \underline{\mathbb{W}}_\alpha, \\ \mathbb{Y} &= \sum_{\alpha=1}^{18} \mathbb{Y}_\alpha \underline{\mathbb{W}}_\alpha = \sum_{\alpha=1}^{18} (\underline{\mathbf{u}}_\alpha \overset{2}{\otimes} \underset{\sim}{\mathbf{P}} + \underline{\mathbf{v}}_\alpha \overset{2}{\otimes} \underset{\sim}{\boldsymbol{\mu}}) \underline{\mathbb{W}}_\alpha, \end{aligned} \tag{16}$$

where the second formula of (16) is obtained by analogy with the first. It should be noted that the formulas in Equation (16) are decompositions of the tensor columns $\mathbb{X}$ and $\mathbb{Y}$ with respect to the orthonormal basis $\underline{\mathbb{W}}_\alpha$, $\alpha = \overline{1,18}$, and $\mathbb{X}_\alpha$ and $\mathbb{Y}_\alpha$ are projections $\mathbb{X}$ and $\mathbb{Y}$ respectively on $\underline{\mathbb{W}}_\alpha$.

It is not difficult to see that, taking into account the first equality of (15), from the second relation of (13) we obtain the following presentations of stress and couple-stress tensors:

$$\begin{aligned} \underset{\sim}{\mathbf{P}} &= \sum_{p=1}^{18} \lambda_p \mathbb{X}_p \underline{\mathbf{u}}_p = \sum_{p=1}^{18} \lambda_p (\underline{\mathbf{u}}_p \overset{2}{\otimes} \underset{\sim}{\boldsymbol{\gamma}} + \underline{\mathbf{v}}_p \overset{2}{\otimes} \underset{\sim}{\boldsymbol{\varkappa}}) \underline{\mathbf{u}}_p, \\ \underset{\sim}{\boldsymbol{\mu}} &= \sum_{p=1}^{18} \lambda_p \mathbb{X}_p \underline{\mathbf{v}}_p = \sum_{p=1}^{18} \lambda_p (\underline{\mathbf{u}}_p \overset{2}{\otimes} \underset{\sim}{\boldsymbol{\gamma}} + \underline{\mathbf{v}}_p \overset{2}{\otimes} \underset{\sim}{\boldsymbol{\varkappa}}) \underline{\mathbf{v}}_p. \end{aligned} \tag{17}$$

It is not difficult to write the reverse CR. In fact, for example, from the second equality of (4) by virtue of (10) and the second equality of (15) we find

$$\begin{aligned} \mathbb{X} &= \underset{\approx}{\mathbb{M}}^{-1} \overset{2}{\otimes} \mathbb{Y} = \sum_{p=1}^{18} \lambda_p^{-1} \underline{\mathbb{W}}_p \underline{\mathbb{W}}_p^T \overset{2}{\otimes} \mathbb{Y} = \sum_{p=1}^{18} \lambda_p^{-1} \mathbb{Y}_p \underline{\mathbb{W}}_p, \\ \mathbb{X}_\alpha &= \lambda_\alpha^{-1} \mathbb{Y}_\alpha, \quad \langle \alpha = \overline{1,18} \rangle. \end{aligned} \tag{18}$$

Taking into account the second equality of (15), from the first equality of (18) similarly to (17) we have

$$\begin{aligned} \underset{\sim}{\boldsymbol{\gamma}} &= \sum_{p=1}^{18} \lambda_p^{-1} \mathbb{Y}_p \underline{\mathbf{u}}_p = \sum_{p=1}^{18} \lambda_p^{-1} (\underline{\mathbf{u}}_p \overset{2}{\otimes} \underset{\sim}{\mathbf{P}} + \underline{\mathbf{v}}_p \overset{2}{\otimes} \underset{\sim}{\boldsymbol{\mu}}) \underline{\mathbf{u}}_p, \\ \underset{\sim}{\boldsymbol{\varkappa}} &= \sum_{p=1}^{18} \lambda_p^{-1} \mathbb{Y}_p \underline{\mathbf{v}}_p = \sum_{p=1}^{18} \lambda_p^{-1} (\underline{\mathbf{u}}_p \overset{2}{\otimes} \underset{\sim}{\mathbf{P}} + \underline{\mathbf{v}}_p \overset{2}{\otimes} \underset{\sim}{\boldsymbol{\mu}}) \underline{\mathbf{v}}_p. \end{aligned}$$

It is not difficult to see that $\mathbb{X}$ can be represented as

$$\mathbb{X} = \begin{pmatrix} \underset{\sim}{\boldsymbol{\gamma}} \\ \underset{\sim}{\boldsymbol{\varkappa}} \end{pmatrix} = \begin{pmatrix} \mathbf{r}^i \underset{\sim}{\mathbf{E}} \nabla_i - \underset{\approx}{\mathbf{C}} \\ \mathbf{r}^i \underset{\sim}{\mathbf{E}} \nabla_i \end{pmatrix} \cdot \begin{pmatrix} \mathbf{u} \\ \boldsymbol{\varphi} \end{pmatrix} = \underset{\approx}{\mathbb{H}} \cdot \mathbb{U}, \quad \underset{\approx}{\mathbb{H}} = \begin{pmatrix} \mathbf{r}^i \underset{\sim}{\mathbf{E}} \nabla_i - \underset{\approx}{\mathbf{C}} \\ \mathbf{r}^i \underset{\sim}{\mathbf{E}} \nabla_i \end{pmatrix}. \tag{19}$$

By virtue of Equation (19), the CR in Equation (4) can also be written in the form

$$\mathbb{Y} = \underset{\approx}{\mathbb{M}} \overset{2}{\otimes} \underset{\approx}{\mathbb{H}} \cdot \mathbb{U}. \tag{20}$$

## 4. On Static Boundary Conditions in the Linear Three-Dimensional Micropolar Theory of Elasticity. Tensor–Operator of Stress and Couple-Stress

Given (2), the static boundary conditions can be written as follows:

$$\mathbf{n} \cdot \underset{\sim}{\mathbf{P}} = \underset{\sim}{\mathbf{T}}^{(1)} \cdot \mathbf{u} + \underset{\sim}{\mathbf{T}}^{(2)} \cdot \boldsymbol{\varphi} = \mathbf{P}, \quad \mathbf{n} \cdot \underset{\sim}{\boldsymbol{\mu}} = \underset{\sim}{\mathbf{T}}^{(3)} \cdot \mathbf{u} + \underset{\sim}{\mathbf{T}}^{(4)} \cdot \boldsymbol{\varphi} = \boldsymbol{\mu}, \tag{21}$$

which by virtue of (20) is still equivalent to equalities

$$
\mathbf{n} \cdot \mathbb{Y} = \begin{pmatrix} \underset{\sim}{\mathbf{T}}^{(1)} & \underset{\sim}{\mathbf{T}}^{(2)} \\ \underset{\sim}{\mathbf{T}}^{(3)} & \underset{\sim}{\mathbf{T}}^{(4)} \end{pmatrix} \cdot \begin{pmatrix} \mathbf{u} \\ \boldsymbol{\varphi} \end{pmatrix} = \mathbf{n} \cdot \underset{\approx}{\mathbb{M}} \overset{2}{\otimes} \underset{\approx}{\mathbb{H}} \cdot \mathbb{U}. \tag{22}
$$

Here **P** and $\boldsymbol{\mu}$ are the stress and couple-stress vectors given on the body surface and the following differential tensors–operators are introduced:

$$
\begin{array}{ll}
\underset{\sim}{\mathbf{T}}^{(1)} = \mathbf{r}_j \mathbf{r}_l n_i A^{ijkl} \nabla_k, & \underset{\sim}{\mathbf{T}}^{(2)} = \mathbf{r}_j \mathbf{r}_l n_i B^{ijkl} \nabla_k - \mathbf{n} \cdot \underset{\approx}{\mathbf{A}} \overset{2}{\otimes} \underset{\sim}{\mathbf{C}}, \\
\underset{\sim}{\mathbf{T}}^{(3)} = \mathbf{r}_j \mathbf{r}_l n_i C^{ijkl} \nabla_k, & \underset{\sim}{\mathbf{T}}^{(4)} = \mathbf{r}_j \mathbf{r}_l n_i D^{ijkl} \nabla_k - \mathbf{n} \cdot \underset{\approx}{\mathbf{C}} \overset{2}{\otimes} \underset{\sim}{\mathbf{C}}.
\end{array} \tag{23}
$$

Introducing the tensor–block matrix operator stress and couple-stress and the vector column of stress and couple-stress vectors

$$
\mathbb{T} = \begin{pmatrix} \underset{\sim}{\mathbf{T}}^{(1)} & \underset{\sim}{\mathbf{T}}^{(2)} \\ \underset{\sim}{\mathbf{T}}^{(3)} & \underset{\sim}{\mathbf{T}}^{(4)} \end{pmatrix} = \mathbf{n} \cdot \underset{\approx}{\mathbb{M}} \overset{2}{\otimes} \underset{\approx}{\mathbb{H}}, \quad \mathbb{Q} = \begin{pmatrix} \mathbf{P} \\ \boldsymbol{\mu} \end{pmatrix}, \tag{24}
$$

the static boundary conditions (21), taking into account the notation given by the second relation of (5), can be written in the form

$$
\mathbb{T} \cdot \mathbb{U} \big|_S = \mathbb{Q}. \tag{25}
$$

We note that the kinematic boundary conditions are represented in the form

$$
\mathbb{U} \big|_S = \mathbb{Z} \quad \left( \mathbb{Z} = \begin{pmatrix} \mathbf{f} \\ \boldsymbol{\psi} \end{pmatrix} \right), \tag{26}
$$

mixed boundary conditions can be written as follows:

$$
\mathbb{T} \cdot \mathbb{U} \big|_{S_1} = \mathbb{Q}; \quad \mathbb{U} \big|_{S_2} = \mathbb{Z}, \tag{27}
$$

and the initial conditions have the form

$$
\mathbb{U} \big|_{t=t_0} = \mathbb{U}_0 \quad \mathbb{V} \big|_{t=t_0} = \mathbb{V}_0, \tag{28}
$$

where

$$
\mathbb{U}_0 = \begin{pmatrix} \mathbf{u}_0 \\ \boldsymbol{\varphi}_0 \end{pmatrix}, \quad \mathbb{V}_0 = \frac{d\mathbb{U}}{dt} \Big|_{t=t_0} = \begin{pmatrix} \mathbf{v}_0 \\ \boldsymbol{\omega}_0 \end{pmatrix}, \quad \mathbf{v}_0 = \frac{d\mathbf{u}}{dt} \Big|_{t=t_0}, \quad \boldsymbol{\omega}_0 = \frac{d\boldsymbol{\varphi}}{dt} \Big|_{t=t_0}.
$$

Here **f** and $\boldsymbol{\psi}$ are the displacement and rotation vectors given on the body surface, $\mathbf{u}_0$ and $\boldsymbol{\varphi}_0$ are the displacement and rotation vectors given at the initial instant of time (for $t = t_0$), $\mathbf{v}_0$ and $\boldsymbol{\omega}_0$ are the velocity and angular velocity vectors given at the initial instant of time, $S$ is the body surface, $S_1 \cup S_2 = S$, $S_1 \cap S_2 = \varnothing$.

## 5. Formulation of Initial-Boundary Value Problems

Let us introduce the definitions.

**Definition 1.** *If the displacement and rotation vectors (kinematic boundary conditions) are given on the body boundary S, then such conditions are called boundary conditions of the first kind, and the problem of the micropolar solids mechanics (SM), using these conditions, and also the initial conditions is called the first initial-boundary value problem.*

In the case under consideration, the first initial-boundary value problem includes: the equations of motion (6), the kinematic boundary conditions (26) and the initial conditions (28).

**Definition 2.** *If static boundary conditions (stress and couple-stress vectors) are given on the body boundary S, then such boundary conditions are called boundary conditions of the second kind, and the problem of micropolar SM using them and initial conditions is called the second initial-boundary value problem.*

In the case under consideration, the second initial-boundary value problem includes: the equations of motion (6), the static boundary conditions (25) and the initial conditions (28).

**Definition 3.** *If kinematic boundary conditions are given on one part of the body boundary $S_1$, and on the remaining part of it $S_2$ are given the static boundary conditions, where $S_1 \cup S_2 = S$, $S_1 \cap S_2 = \varnothing$, then such boundary conditions are called mixed boundary conditions, and the problem of micropolar SM, using them and initial conditions is called the mixed initial-boundary value problem.*

In this case, the mixed (third) initial-boundary value problem includes: the equations of motion (6), the kinematic boundary conditions (26) on one part of the body boundary and the static boundary conditions (25) on the rest of the body boundary and the initial conditions (28) (see also (27)).

Note that, excluding the characteristics of the micropolar theory from the above definitions, we obtain the corresponding definitions for classical SM.

It should be noted that the kinematic boundary conditions and the initial conditions do not need to be split, since they were set in a split form. Hence, for the splitting of the first initial-boundary value problem, it is sufficient to split only the equations of motion, since, as already mentioned in the previous proposition, the kinematic boundary conditions and the initial conditions are split. In this connection, the splitting of the static boundary conditions is of great interest. If the equations of motion (6) and the static boundary conditions (25) can be split under some conditions, then under the same conditions all the initial-boundary value problems formulated above can be split. Hence, it is necessary to establish the conditions under which the equations of motion (6) and the static boundary conditions (25) are split.

## 6. Decomposition of the Equation of Motion in the Case of a Homogeneous Isotropic Micropolar Medium

In this case, as many authors (see, for example, [25,30,31]) consider $\underset{\approx}{\mathbf{B}} = 0$ and differential tensors–operators $\underset{\sim}{\mathbf{A}}$, $\underset{\sim}{\mathbf{B}}$, $\underset{\sim}{\mathbf{C}}$ and $\underset{\sim}{\mathbf{D}}$ (see (7)) have the form

$$\underset{\sim}{\mathbf{A}} = \underset{\sim}{\mathbf{E}}Q_2 + d\nabla\nabla, \quad \underset{\sim}{\mathbf{B}} = \underset{\sim}{\mathbf{C}} = -2\alpha \underset{\sim}{\mathbf{C}} \cdot \nabla, \quad \underset{\sim}{\mathbf{D}} = \underset{\sim}{\mathbf{E}}Q_4 + m\nabla\nabla, \quad \underset{\sim}{\mathbf{J}} = J\underset{\sim}{\mathbf{E}},$$
$$Q_2 = b\Delta - \rho\partial_t^2, \quad Q_4 = g\Delta - l - J\partial_t^2, \quad Q_1 = Q_2 + d\Delta, \quad Q_3 = Q_4 + m\Delta,$$
$$d = \lambda + \mu - \alpha, \quad l = 4\alpha, \quad b = \mu + \alpha, \quad g = \delta + \beta, \quad m = \gamma + \delta - \beta,$$

where $Q_1$, $Q_2$, $Q_3$ and $Q_4$ are wave operators, and the elasticity tensors have expressions

$$\underset{\approx}{\mathbf{A}} = a_1\underset{\approx}{\mathbf{C}}_{(1)} + a_2\underset{\approx}{\mathbf{C}}_{(2)} + a_3\underset{\approx}{\mathbf{C}}_{(3)}, \quad \underset{\approx}{\mathbf{D}} = d_1\underset{\approx}{\mathbf{C}}_{(1)} + d_2\underset{\approx}{\mathbf{C}}_{(2)} + d_3\underset{\approx}{\mathbf{C}}_{(3)}. \tag{29}$$

Here $\underset{\approx}{\mathbf{C}}_{(1)}$, $\underset{\approx}{\mathbf{C}}_{(2)}$ and $\underset{\approx}{\mathbf{C}}_{(3)}$ are the basic isotropic tensors of the fourth rank, and for material constants we use the notation $a_1 = \lambda$, $a_2 = \mu$, $a_3 = \alpha$, $b_1 = \gamma$, $b_2 = \delta$ and $b_3 = \beta$.

Denoting by

$$\mathbb{M}_* = \begin{pmatrix} \hat{\underset{\sim}{\mathbf{A}}} & \hat{\underset{\sim}{\mathbf{B}}}^{(1)} \\ \hat{\underset{\sim}{\mathbf{B}}}^{(2)} & \hat{\underset{\sim}{\mathbf{C}}} \end{pmatrix} \tag{30}$$

the tensor–block matrix operator of the cofactors for the tensor–block matrix operator $\mathbb{M}$ of Equation (6), after cumbersome calculations we obtain [27,28,32]

$$
\begin{aligned}
&\hat{\mathbf{A}} = Q_3 P \underline{\mathbf{R}}, \quad \hat{\underline{\mathbf{B}}} = \hat{\underline{\mathbf{B}}}^{(1)} = \hat{\underline{\mathbf{B}}}^{(2)} = Q_1 Q_3 P \underline{\mathbf{B}} \quad (\hat{\underline{\mathbf{B}}}^T = -\hat{\underline{\mathbf{B}}}),\\
&\hat{\underline{\mathbf{C}}} = Q_1 P \underline{\mathbf{T}}; \quad \underline{\mathbf{R}} = \underline{\mathbf{E}} Q_1 Q_4 - (dQ_4 - 4\alpha^2)\nabla\nabla, \quad \underline{\mathbf{B}} = -2\alpha \underline{\mathbf{C}} \cdot \nabla,\\
&\underline{\mathbf{T}} = \underline{\mathbf{E}} Q_2 Q_3 - (mQ_2 - 4\alpha^2)\nabla\nabla, \quad P = Q_2 Q_4 + 4\alpha^2 \Delta.
\end{aligned}
\tag{31}
$$

It is easy to see that by virtue of Equation (31) the tensor–block matrix operator of the cofactors (30) can be represented as follows:

$$
\mathbb{M}_* = P \begin{pmatrix} Q_3 \hat{\underline{\mathbf{R}}} & Q_1 Q_3 \underline{\mathbf{B}} \\ Q_1 Q_3 \underline{\mathbf{B}} & Q_1 \hat{\underline{\mathbf{T}}} \end{pmatrix} = \begin{pmatrix} PQ_3 & 0 \\ 0 & PQ_1 \end{pmatrix} \mathbb{N}^{(1)} = \mathbb{N}^{(2)} \begin{pmatrix} PQ_3 & 0 \\ 0 & PQ_1 \end{pmatrix},
$$

where tensor–block matrix operators are introduced

$$
\mathbb{N}^{(1)} = \begin{pmatrix} \underline{\mathbf{R}} & Q_1 \underline{\mathbf{B}} \\ Q_3 \underline{\mathbf{B}} & \underline{\mathbf{T}} \end{pmatrix}, \quad \mathbb{N}^{(2)} = \begin{pmatrix} \underline{\mathbf{R}} & Q_3 \underline{\mathbf{B}} \\ Q_1 \underline{\mathbf{B}} & \underline{\mathbf{T}} \end{pmatrix}.
$$

It is easy to prove the relations

$$
\mathbb{M} \cdot \mathbb{N}^{(1)T} = \mathbb{N}^{(2)T} \cdot \mathbb{M} = \begin{pmatrix} \underline{\mathbf{E}} Q_1 P & \underline{\mathbf{O}} \\ \underline{\mathbf{O}} & \underline{\mathbf{E}} Q_3 P \end{pmatrix}, \quad \det(\mathbb{M}) = Q_1 Q_3 P^2.
\tag{32}
$$

Then if we shall seek the solution of Equation (6) in the form (similar to Galerkin)

$$
\mathbb{U} = \mathbb{N}^{(1)T} \cdot \mathbb{V} \quad \left( \mathbb{U} = \begin{pmatrix} \mathbf{u} \\ \boldsymbol{\varphi} \end{pmatrix}, \quad \mathbb{V} = \begin{pmatrix} \mathbf{v} \\ \boldsymbol{\psi} \end{pmatrix} \right),
$$

then, by virtue of the appropriate relation (32), we obtain the following split equations:

$$
Q_1 (Q_2 Q_4 + 4\alpha^2 \Delta)\mathbf{v} + \rho\mathbf{F} = 0, \quad Q_3 (Q_2 Q_4 + 4\alpha^2 \Delta)\boldsymbol{\psi} + \rho\mathbf{m} = 0.
\tag{33}
$$

Note that similar equations were obtained by [26]. Similar equations in another way were obtained by [22]. Finally, Sandru and Nowacki gave the same representations of the displacement and rotation vectors and they are reduced to (33). The paper [24] deserve great attention because the system of equilibrium equations for the isotropic elastic body without a center of symmetry and in the absence of mass loads is decomposed into two independent systems of equations.

Applying the operator $\mathbb{N}^{(2)T}$ to the left of (6), by the first relation of (32) we will have

$$
\begin{aligned}
&Q_1 [(Q_2 Q_4 + 4\alpha^2 \Delta)\mathbf{u} + 2\alpha(\underline{\mathbf{C}} \cdot \nabla)\cdot(\rho\mathbf{m})] + \underline{\mathbf{R}} \cdot (\rho\mathbf{F}) = 0,\\
&Q_3 [(Q_2 Q_4 + 4\alpha^2 \Delta)\boldsymbol{\varphi} + 2\alpha(\underline{\mathbf{C}} \cdot \nabla)\cdot(\rho\mathbf{F})] + \underline{\mathbf{T}} \cdot (\rho\mathbf{m}) = 0.
\end{aligned}
\tag{34}
$$

For $\alpha = 0$ (the case of a reduced medium), the classical equation follows from the first equation of (34), and the second equation has a similar form.

## 7. Decomposition of Static Boundary Conditions

In the case of an isotropic micropolar material without a center of symmetry, by virtue of Equation (29) and $\underset{\approx}{\mathbf{B}} = b_1 \underset{\approx}{\mathbf{C}}_{(1)} + b_2 \underset{\approx}{\mathbf{C}}_{(2)} + b_3 \underset{\approx}{\mathbf{C}}_{(3)}$, from Equation (23) we have

$$\underset{\sim}{\mathbf{T}}^{(1)} = a_2 \underset{\sim}{\mathbf{E}} \mathbf{n} \cdot \nabla + a_1 \mathbf{n} \nabla + a_3 (\mathbf{n} \nabla)^T,$$

$$\underset{\sim}{\mathbf{T}}^{(2)} = b_2 \underset{\sim}{\mathbf{E}} \mathbf{n} \cdot \nabla + b_1 \mathbf{n} \nabla + b_3 (\mathbf{n} \nabla)^T - (a_2 - a_3) \mathbf{n} \cdot \underset{\simeq}{\mathbf{C}},$$

$$\underset{\sim}{\mathbf{T}}^{(3)} = b_2 \underset{\sim}{\mathbf{E}} \mathbf{n} \cdot \nabla + b_1 \mathbf{n} \nabla + b_3 (\mathbf{n} \nabla)^T,$$

$$\underset{\sim}{\mathbf{T}}^{(4)} = d_2 \underset{\sim}{\mathbf{E}} \mathbf{n} \cdot \nabla + d_1 \mathbf{n} \nabla + d_3 (\mathbf{n} \nabla)^T - (b_2 - b_3) \mathbf{n} \cdot \underset{\simeq}{\mathbf{C}}.$$

It should be noted that some authors (see, for example, [25,30,31]) consider that $\underset{\approx}{\mathbf{B}}$ is an asymmetric tensor, therefore in the case of an isotropic medium it is zero, as was done above. However, some authors prove that $\underset{\approx}{\mathbf{B}}$ is a symmetric tensor, therefore, in the case of an isotropic medium, it is not equal to zero, and how any isotropic fourth-rank tensor is generally determined by three parameters, as is customary in this case. Further it is easy to see that

$$\underset{\sim}{\mathbf{T}}^{(2)} = \underset{\sim}{\mathbf{T}}^{(3)} - (a_2 - a_3) \mathbf{n} \cdot \underset{\simeq}{\mathbf{C}}, \quad \underset{\sim}{\mathbf{T}}^{(4)} = \underset{\sim}{\mathbf{T}}^{'(4)} - (b_2 - b_3) \mathbf{n} \cdot \underset{\simeq}{\mathbf{C}},$$

$$\underset{\sim}{\mathbf{T}}^{'(4)} = d_2 \underset{\sim}{\mathbf{E}} \mathbf{n} \cdot \nabla + d_1 \mathbf{n} \nabla + d_3 (\mathbf{n} \nabla)^T.$$

Assuming that the body has a piecewise-plane boundary and denoting by $\underset{\sim}{\mathbf{T}}_*^{(1)}$ and $|\underset{\sim}{\mathbf{T}}^{(1)}|$, $\underset{\sim}{\mathbf{T}}_*^{(3)}$ and $|\underset{\sim}{\mathbf{T}}^{(3)}|$, $\underset{\sim}{\mathbf{T}}_*^{'(4)}$ and $|\underset{\sim}{\mathbf{T}}^{'(4)}|$ the differential tensors–operators of the cofactors and determinants for the tensor–operators $\underset{\sim}{\mathbf{T}}^{(1)}$, $\underset{\sim}{\mathbf{T}}^{(3)}$ and $\underset{\sim}{\mathbf{T}}^{'(4)}$ respectively, after simple calculations we obtain

$$\underset{\sim}{\mathbf{T}}_*^{(1)} = [(a_1 + a_2)(a_2 + a_3) \underset{\sim}{\mathbf{E}} \mathbf{n} \cdot \nabla - a_3 (a_1 + a_2) \mathbf{n} \nabla$$
$$- a_1 (a_2 + a_3)(\mathbf{n} \nabla)^T] \mathbf{n} \cdot \nabla + a_1 a_3 [\nabla \nabla + (\mathbf{n} \mathbf{n} - \underset{\sim}{\mathbf{E}}) \Delta],$$

$$|\underset{\sim}{\mathbf{T}}^{(1)}| = a_2 [(a_1 + a_2)(a_2 + a_3) \mathbf{n} \mathbf{n} \overset{2}{\otimes} \nabla \nabla - a_1 a_3 \Delta] \mathbf{n} \cdot \nabla,$$

$$\underset{\sim}{\mathbf{T}}_*^{(3)} = [(b_1 + b_2)(b_2 + b_3) \underset{\sim}{\mathbf{E}} \mathbf{n} \cdot \nabla - b_3 (b_1 + b_2) \mathbf{n} \nabla$$
$$- b_1 (b_2 + b_3)(\mathbf{n} \nabla)^T] \mathbf{n} \cdot \nabla + b_1 b_3 [\nabla \nabla + (\mathbf{n} \mathbf{n} - \underset{\sim}{\mathbf{E}}) \Delta],$$

$$|\underset{\sim}{\mathbf{T}}^{(3)}| = b_2 [(b_1 + b_2)(b_2 + b_3) \mathbf{n} \mathbf{n} \overset{2}{\otimes} \nabla \nabla - b_1 b_3 \Delta] \mathbf{n} \cdot \nabla,$$

$$\underset{\sim}{\mathbf{T}}_*^{'(4)} = [(d_1 + d_2)(d_2 + d_3) \underset{\sim}{\mathbf{E}} \mathbf{n} \cdot \nabla - d_3 (d_1 + d_2) \mathbf{n} \nabla$$
$$- d_1 (d_2 + d_3)(\mathbf{n} \nabla)^T] \mathbf{n} \cdot \nabla + d_1 d_3 [\nabla \nabla + (\mathbf{n} \mathbf{n} - \underset{\sim}{\mathbf{E}}) \Delta],$$

$$|\underset{\sim}{\mathbf{T}}^{'(4)}| = d_2 [(d_1 + d_2)(d_2 + d_3) \mathbf{n} \mathbf{n} \overset{2}{\otimes} \nabla \nabla - d_1 d_3 \Delta] \mathbf{n} \cdot \nabla.$$

Note that we want to obtain boundary conditions separately for $\mathbf{u}$ and $\boldsymbol{\varphi}$. In order to shorten the letter, we consider the case when $b_2 = b_3$, $a_2 = a_3$. Then $\underset{\sim}{\mathbf{T}}^{(2)} = \underset{\sim}{\mathbf{T}}^{(3)}$, $\underset{\sim}{\mathbf{T}}^{(4)} = \underset{\sim}{\mathbf{T}}^{'(4)}$ and the boundary conditions from Equation (25) can be written in the form

$$\underset{\sim}{\mathbf{T}}^{(1)} \cdot \mathbf{u} + \underset{\sim}{\mathbf{T}}^{(3)} \cdot \boldsymbol{\varphi} = \mathbf{P}, \quad \underset{\sim}{\mathbf{T}}^{(3)} \cdot \mathbf{u} + \underset{\sim}{\mathbf{T}}^{'(4)} \cdot \boldsymbol{\varphi} = \boldsymbol{\mu}.$$

In this case, it is easy to obtain the boundary conditions separately for **u** and $\boldsymbol{\varphi}$

$$\left( |\mathbf{T}'^{(4)}| \mathbf{T}_*^{(3)T} \cdot \mathbf{T}^{(1)} - |\mathbf{T}^{(3)}| \mathbf{T}'^{(4)T}_* \cdot \mathbf{T}^{(3)} \right) \cdot \mathbf{u}$$

$$= |\mathbf{T}'^{(4)}| \mathbf{T}_*^{(3)T} \cdot \mathbf{P} - |\mathbf{T}^{(3)}| \mathbf{T}'^{(4)T}_* \cdot \boldsymbol{\mu},$$

$$\left( |\mathbf{T}^{(3)}| \mathbf{T}_*^{(1)T} \cdot \mathbf{T}^{(3)} - |\mathbf{T}^{(1)}| \mathbf{T}_*^{(3)T} \cdot \mathbf{T}'^{(4)} \right) \cdot \boldsymbol{\varphi}$$

$$= |\mathbf{T}^{(3)}| \mathbf{T}_*^{(1)T} \cdot \mathbf{P} - |\mathbf{T}^{(1)}| \mathbf{T}_*^{(3)T} \cdot \boldsymbol{\mu}.$$

Note that the eigenvalue problem can also be considered for differential tensor–operator and tensor–block matrix operator. Consequently, in this case the eigenvalues are differential operators whose product, taking into account their multiplicities, will be equal to the determinant of the object for which the eigenvalue problem is considered. Of course, we can find our eigenoperators from the solution of the characteristic equation. For example, the characteristic equation for an arbitrary tensor–block matrix $\mathbb{M}$ of the second rank consisting of four second-rank tensors of three-dimensional space has the form [29,34]

$$\eta^6 - I_1(\mathbb{M})\eta^5 + I_2(\mathbb{M})\eta^4 - I_3(\mathbb{M})\eta^3 + I_4(\mathbb{M})\eta^2 - I_5(\mathbb{M})\eta + I_6(\mathbb{M}) = 0, \tag{35}$$

$$S_k = I_k(\mathbb{M}) = \frac{1}{k!} \begin{vmatrix} s_1 & 1 & \cdots & 0 & 0 \\ s_2 & s_1 & \cdots & 0 & 0 \\ \cdots & \cdots & \cdots & \cdots & \cdots \\ s_{k-1} & s_{k-2} & \cdots & s_1 & k \\ s_k & s_{k-1} & \cdots & s_2 & s_1 \end{vmatrix}, \quad k = \overline{1,6}, \tag{36}$$

$$S_k = I_k(\mathbb{M}), \quad s_k = I_1(\mathbb{M}^k), \quad k = \overline{1,6}, \quad \mathbb{M}^k = \overbrace{\mathbb{M} \cdot \mathbb{M} \cdot \ldots \cdot \mathbb{M}}^{k},$$

where $I_k(\mathbb{M}), k = \overline{1,6}$, denote the invariants of the tensor–block matrix $\mathbb{M}$. In this case, the inverse relations to Equation (36) are represented in the form

$$s_k = I_1(\mathbb{M}^k) = \begin{vmatrix} S_1 & 1 & 0 & \cdots & 0 \\ 2S_2 & S_1 & 1 & \cdots & 0 \\ \cdots & \cdots & \cdots & \cdots & \cdots \\ kS_k & S_{k-1} & S_{k-2} & \cdots & S_1 \end{vmatrix}, \quad k = \overline{1,6}. \tag{37}$$

Replacing in Equations (35)–(37) $\mathbb{M}$ by the tensor–block matrix operators of the equations of motion in displacements and rotations under various anisotropic media, we obtain the characteristic equations for them. If we find the roots (eigenoperators) of the obtained characteristic equations for the above tensor–block matrix operators, then their determinants can be represented as a product of simple eigenoperators. Thus, the equations are split.

## 8. Decomposition of the Canonical Equations of the Classical Elasticity Theory for the Transversely Isotropic Body

Under the canonical equations we call equations that are obtained by the canonical presentations of the material tensors. In a similar sense, the term canonical mechanics can also be used. Thus, in this case the elastic tensor $\underset{\approx}{\mathbf{A}}$ is represented in the canonical form [29,34], that is

$$\underset{\approx}{\mathbf{A}} = \sum_{k=1}^{6} \lambda_k \mathbf{w}_k \mathbf{w}_k, \tag{38}$$

where $\lambda_k$ and $\underset{\sim}{\mathbf{w}}_k$, $k = 1, \ldots, 6$, are the eigenvalues and eigentensors for $\underset{\approx}{\mathbf{A}}$. Then by Equation (38) the vector equation with respect to the displacement vector can be written in the form

$$\underset{\sim}{\mathbf{L}} \cdot \mathbf{u} + \rho \mathbf{F} = 0, \quad \underset{\sim}{\mathbf{L}} = \sum_{k=1}^{6} \lambda_k \underset{\sim}{\mathbf{w}}_k \cdot \nabla \underset{\sim}{\mathbf{w}}_k \cdot \nabla. \tag{39}$$

It is seen that for the splitting of the last equation it is necessary to find the cofactors $\underset{\sim}{\mathbf{L}}_*$ for $\underset{\sim}{\mathbf{L}}$ (see the second relation of (39)), and for this, in turn it is necessary to find expressions for the determinant of a linear combination of several tensors of the second rank (from two to six). We give below the expressions for the determinant and the cofactors of the sum of six second-rank tensors without proofs, from which, in turn, it is easy to obtain analogous expressions for the sum of a smaller number of tensors. The expressions for the determinant and the cofactor tensor of the sum of six tensors of the second rank are presented as follows:

$$| \sum_{k=1}^{6} \mathbf{A}_k | = \sum_{k=1}^{6} |\mathbf{A}_k| + \sum_{i=1}^{6} I_2(\mathbf{A}_i) \big[ \delta_i \sum_{j=1}^{6} I_1(\mathbf{A}_j) - I_1(\mathbf{A}_i) \big]$$

$$+ \prod_{i=1}^{3} \big[ I_1(\mathbf{A}_{2i-1}) + I_1(\mathbf{A}_{2i}) \big] + \sum_{i=1}^{3} I_1(\mathbf{A}_{2i-1}) I_1(\mathbf{A}_{2i}) \big[ \delta_i \sum_{k=1}^{6} I_1(\mathbf{A}_k) - I_1(\mathbf{A}_{2i-1})$$

$$- I_1(\mathbf{A}_{2i}) \big] + \sum_{i=1}^{6} \big[ \sum_{j=1}^{6} I_1(\mathbf{A}_i^2 \cdot \mathbf{A}_j) - I_1(\mathbf{A}_i^3) \big] + \sum_{i=1}^{4} \sum_{j=i+1}^{5} \sum_{k=j+1}^{6} I_1 \big[ \mathbf{A}_i \cdot (\mathbf{A}_j \cdot \mathbf{A}_k$$

$$+ \mathbf{A}_k \cdot \mathbf{A}_j) \big] - \big[ \sum_{i=1}^{6} I_1(\mathbf{A}_i) \big] \big[ \sum_{j=1}^{5} \sum_{k=j+1}^{6} I_1(\mathbf{A}_j \cdot \mathbf{A}_k), \quad \delta_i = \sum_{j=1}^{6} \delta_{ij}, \quad (i = \overline{1,6});$$

$$\big( \sum_{i=1}^{6} \mathbf{A}_i \big)_* = \sum_{i=1}^{6} \mathbf{A}_{i*} + \sum_{i=1}^{5} \sum_{j=i+1}^{6} {}^{4}\mathbf{A}_{i*} \cdot \mathbf{A}_j; \quad \mathbf{A}_{i*} \equiv \frac{\partial |\mathbf{A}_i|}{\partial \mathbf{A}_i}, \quad {}^{4}\mathbf{A}_{i*} \equiv \frac{\partial^2 |\mathbf{A}_i|}{\partial \mathbf{A}_i \partial \mathbf{A}_i}.$$

Given $I_1({}^{4}\mathbf{A}_{i*} \cdot \mathbf{A}_j) = I_1(\mathbf{A}_i) I_1(\mathbf{A}_j) - I_1(\mathbf{A}_i \cdot \mathbf{A}_j)$, we obtain from the previous formula

$$I_1 \big[ \big( \sum_{i=1}^{6} \mathbf{A}_i \big)_* \big] = I_2 \big( \sum_{i=1}^{6} \mathbf{A}_i \big) = \sum_{i=1}^{6} I_2(\mathbf{A}_i) + \sum_{i=1}^{5} \sum_{j=i+1}^{6} \big[ I_1(\mathbf{A}_i) I_1(\mathbf{A}_j) - I_1(\mathbf{A}_i \cdot \mathbf{A}_j) \big].$$

Here $\mathbf{A}_{i*}$ and ${}^{4}\mathbf{A}_{i*}$, $i = \overline{1,6}$, are tensors of cofactors of second rank and second order and fourth rank and first order, respectively.

Further, before we consider the canonical presentations of tensors, we introduce the following definition.

**Definition 4.** *The symbol $\{\alpha_1, \alpha_2, \ldots, \alpha_k\}$, where k is the number of different eigenvalues of the tensor, and $\alpha_i$ is the multiplicity of the eigenvalue $\lambda_i$ ($i = 1, 2, \ldots, k$), is called the anisotropy symbol (structure symbol) of the tensor.*

We note that on the basis of this definition, the classification of classical and microcontinuum anisotropic materials is given in [29,34]. By virtue of this classification, classical (micropolar) isotropic materials are special cases of materials in which the anisotropy symbol consists of not more than two (three) elements. A similar situation occurs for other anisotropies [29,34]. In particular, classical transversely isotropic materials are special cases of anisotropic materials whose anisotropy symbols consist of four elements, and orthotropic materials are special cases of anisotropic media whose structure symbols consist of no more than 6 elements. The anisotropy symbol of an orthotropic micropolar material with a symmetry center consists of not more than nine elements [29,34]. It should also be noted that the mathematical structure of elastic media was investigated in [13], and the classification of classical elastic media is given in [9].

We now consider the canonical representation of the transversely isotropic elastic modulus tensor with the anisotropy symbol {1,1,2,2} [29,34]:

$$\underset{\approx}{\mathbf{A}} = \mu_1\underset{\sim}{\mathbf{w}}_1\underset{\sim}{\mathbf{w}}_1 + \mu_2\underset{\sim}{\mathbf{w}}_2\underset{\sim}{\mathbf{w}}_2 + \mu_3(\underset{\sim}{\mathbf{w}}_3\underset{\sim}{\mathbf{w}}_3 + \underset{\sim}{\mathbf{w}}_4\underset{\sim}{\mathbf{w}}_4) + \mu_5(\underset{\sim}{\mathbf{w}}_5\underset{\sim}{\mathbf{w}}_5 + \underset{\sim}{\mathbf{w}}_6\underset{\sim}{\mathbf{w}}_6). \tag{40}$$

(by force of the classification of materials adopted in [29,34], transversely isotropic materials can be of the following types: {1,1,2,2}, {1,2,1,2}, {1,2,2,1}, {2,1,1,2}, {2,1,2,1}, {2,2,1,1}). For the material under consideration, the eigenvalues are determined by the formulas [29,34]

$$\mu_1 = 1/2(A_{11} + A_{12} + A_{33} - \sqrt{(A_{11} + A_{12} - A_{33})^2 + 8A_{13}^2}),$$

$$\mu_2 = 1/2(A_{11} + A_{12} + A_{33} + \sqrt{(A_{11} + A_{12} - A_{33})^2 + 8A_{13}^2}), \tag{41}$$

$$\mu_3 = \mu_4 = A_{11} - A_{12}, \quad \mu_5 = \mu_6 = A_{55},$$

and the eigentensors are represented in the form

$$\underset{\sim}{\mathbf{w}}_1 = -\frac{\sqrt{2}}{2}\sin\alpha(\underset{\sim}{\mathbf{e}}_1 + \underset{\sim}{\mathbf{e}}_2) + \cos\alpha\underset{\sim}{\mathbf{e}}_3 = -\frac{\sqrt{2}}{2}\sin\alpha\underset{\sim}{\mathbf{I}} + \cos\alpha\underset{\sim}{\mathbf{e}}_3,$$

$$\underset{\sim}{\mathbf{w}}_2 = \frac{\sqrt{2}}{2}\cos\alpha(\underset{\sim}{\mathbf{e}}_1 + \underset{\sim}{\mathbf{e}}_2) + \sin\alpha\underset{\sim}{\mathbf{e}}_3 = \frac{\sqrt{2}}{2}\cos\alpha\underset{\sim}{\mathbf{I}} + \sin\alpha\underset{\sim}{\mathbf{e}}_3, \tag{42}$$

$$\underset{\sim}{\mathbf{w}}_3 = \frac{\sqrt{2}}{2}(\underset{\sim}{\mathbf{e}}_1 - \underset{\sim}{\mathbf{e}}_2), \quad \underset{\sim}{\mathbf{w}}_4 = \underset{\sim}{\mathbf{e}}_4, \quad \underset{\sim}{\mathbf{w}}_5 = \underset{\sim}{\mathbf{e}}_5, \quad \underset{\sim}{\mathbf{w}}_6 = \underset{\sim}{\mathbf{e}}_6;$$

$$\underset{\sim}{\mathbf{e}}_\alpha = \mathbf{e}_\alpha\mathbf{e}_\alpha, \quad \alpha = 1,2,3, \quad \underset{\sim}{\mathbf{e}}_4 = \frac{1}{\sqrt{2}}(\mathbf{e}_3\mathbf{e}_2 + \mathbf{e}_2\mathbf{e}_3), \quad \underset{\sim}{\mathbf{e}}_5 = \frac{1}{\sqrt{2}}(\mathbf{e}_1\mathbf{e}_3 + \mathbf{e}_3\mathbf{e}_1),$$

$$\underset{\sim}{\mathbf{e}}_6 = \frac{1}{\sqrt{2}}(\mathbf{e}_2\mathbf{e}_1 + \mathbf{e}_1\mathbf{e}_2), \quad \mathbf{e}_i \cdot \mathbf{e}_j = \delta_{ij}, \quad i,j = 1,2,3, \quad \text{tg}2\alpha = \frac{2\sqrt{2}A_{13}}{A_{11} + A_{12} - A_{33}},$$

$$\underset{\sim}{\mathbf{e}}_m \overset{2}{\otimes} \underset{\sim}{\mathbf{e}}_n = \delta_{mn}, \quad m,n = \overline{1,6}, \quad \underset{\sim}{\mathbf{I}} = \mathbf{e}_1\mathbf{e}_1 + \mathbf{e}_2\mathbf{e}_2 = \underset{\sim}{\mathbf{e}}_1 + \underset{\sim}{\mathbf{e}}_2.$$

The canonical representations of the tensor–operator of the equations and its determinant and the tensor of cofactors and its components, as well as the stress tensor–operator, its determinant and the components of the tensor–operator of cofactors by virtue of Equations (40)–(42) have the form

$$\underset{\sim}{\mathbf{L}} = (a_1\underset{\sim}{\mathbf{I}} + a_3\mathbf{e}_3\mathbf{e}_3)\Delta + a_2\nabla^0\nabla^0 + a_5[\mathbf{e}_3\nabla^0 + (\mathbf{e}_3\nabla^0)^T]\partial_3 + (a_3\underset{\sim}{\mathbf{I}} + a_4\mathbf{e}_3\mathbf{e}_3)\partial_3^2;$$

$$|\underset{\sim}{\mathbf{L}}| = A\Delta^3 + B\Delta^2\partial_3^2 + C\Delta\partial_3^4 + D\partial_3^6 = k(\Delta + k_1\partial_3^2)(\Delta + k_2\partial_3^2)(\Delta + k_3\partial_3^2),$$

$$k = A, \quad k_1 + k_2 + k_3 = B/A, \quad k_1k_2 + k_1k_3 + k_2k_3 = C/A, \quad k_1k_2k_3 = D/A;$$

$$\underset{\sim}{\mathbf{L}}_* = a(a_3\underset{\sim}{\mathbf{I}} + a_1\mathbf{e}_3\mathbf{e}_3)\Delta^2 - a_2a_3\Delta\nabla^0\nabla^0 - a_1a_5[\mathbf{e}_3\nabla^0 + (\mathbf{e}_3\nabla^0)^T]\partial_3$$

$$+ \left\{\left[[(a_2 + a_3)a_4 + c(a_3 + a_5)]\underset{\sim}{\mathbf{I}} + a_3(a + a_1)\mathbf{e}_3\mathbf{e}_3\right]\Delta + (a_5^2 - a_2a_4)\nabla^0\nabla^0\right\}\partial_3^2$$

$$- a_3a_5[\mathbf{e}_3\nabla^0 + (\mathbf{e}_3\nabla^0)^T]\partial_3^3 - a_3(a_4\underset{\sim}{\mathbf{I}} - a_3\mathbf{e}_3\mathbf{e}_3)\partial_3^4;$$

$$\underset{\sim}{\mathbf{T}} = (a_1\underset{\sim}{\mathbf{I}} + a_3\mathbf{e}_3\mathbf{e}_3)\mathbf{n}^0 \cdot \nabla^0 - b\mathbf{n}^0\nabla^0 + a_1(\mathbf{n}^0\nabla^0)^T + n_3[-c\mathbf{e}_3\nabla^0 + a_3(\mathbf{e}_3\nabla^0)^T]$$

$$+ [n_3(a_3\underset{\sim}{\mathbf{I}} + a_4\mathbf{e}_3\mathbf{e}_3) - c\mathbf{n}^0\mathbf{e}_3 + a_3\mathbf{e}_3\mathbf{n}^0]\partial_3;$$

$$|\underset{\sim}{\mathbf{T}}| = a_1a_3\{[b - (b+c)n_3^2]\Delta + 2a_2\mathbf{n}^0\mathbf{n}^0\overset{2}{\otimes}\nabla^0\nabla^0\}\mathbf{n}^0 \cdot \nabla^0$$

$$+ \{b(a_1a_4 - a_3^2) + a_1c^2 - [(c-b)a_3^2 + a(c^2 + ba_4)]n_3^2\}n_3\Delta\partial_3$$

$$+ 2a_1[(a_3 + c)a_5 + a_2a_4]n_3\mathbf{n}^0\mathbf{n}^0\overset{2}{\otimes}\nabla^0\nabla^0\partial_3 + a_3\{-a_1c$$

$$+ [a_1c + (a+a_1)a_4 - c^2]n_3^2\}\mathbf{n}^0 \cdot \nabla^0\partial_3^2 - a_3^2[c - (c+a_4)n_3^2]n_3\partial_3^3;$$

$$\tag{43}$$

$$
\underset{\sim}{\mathbf{T}}_* = a_3 \mathbf{n}^0 \cdot \nabla^0 [a \underset{\sim}{\mathbf{I}}\, \mathbf{n}^0 \cdot \nabla^0 + b \mathbf{n}^0 \nabla^0 - a_1 (\mathbf{n}^0 \nabla^0)^T] + c a_3 n_3^2 (\mathbf{I}\Delta - \nabla^0 \nabla^0)
$$

$$
+\frac{1}{2}\{[(3a_1+a_2)a_4-c^2]\underset{\sim}{\mathbf{I}}\mathbf{n}^0\cdot\nabla^0+(ba_4+c^2)\mathbf{n}^0\nabla^0-(2a_1a_4-a_3^2)(\mathbf{n}^0\nabla^0)^T\}n_3\partial_3
$$

$$
+a_3[(a_4 n_3^2+c|\mathbf{n}^0|^2)\underset{\sim}{\mathbf{I}}-c\mathbf{n}^0\mathbf{n}^0]\partial_3^2 - \underset{\sim}{\mathbf{C}}\overset{2}{\otimes}\mathbf{n}^0\nabla^0\{ba_3[n_3\underset{\sim}{\mathbf{C}}\cdot(\mathbf{e}_3\nabla^0)^T-\mathbf{e}_3\underset{\sim}{\mathbf{C}}\cdot\mathbf{n}^0\partial_3
$$

$$
-a_1 c(\underset{\sim}{\mathbf{C}}\cdot\mathbf{n}^0\mathbf{e}_3\partial_3-n_3\mathbf{e}_3\underset{\sim}{\mathbf{C}}\cdot\nabla^0)\} - a_1\mathbf{n}^0\cdot\nabla^0\{a_3[n_3(\mathbf{e}_3\nabla^0)^T+\mathbf{e}_3\mathbf{n}^0\partial_3]
$$

$$
-c(\mathbf{n}^0\mathbf{e}_3\partial_3+n_3\mathbf{e}_3\nabla^0)\}-a_3^2[n_3^2(\mathbf{e}_3\nabla^0)^T\partial_3+n_3\mathbf{e}_3\mathbf{n}^0\partial_3^2]
$$

$$
+ca_3(n_3^2\mathbf{e}_3\nabla^0\partial_3+n_3\mathbf{n}^0\mathbf{e}_3\partial_3^2)+\mathbf{e}_3\mathbf{e}_3[a_1 b|\mathbf{n}^0|^2\Delta+2a_1 a_2\mathbf{n}^0\mathbf{n}^0\overset{2}{\otimes}\nabla^0\nabla^0
$$

$$
+(a_1 a_5+a_2 a_3)n_3\mathbf{n}^0\cdot\nabla^0\partial_3+a_3^2 n_3^2\partial_3^2],
$$

where the following notations are introduced:

$$
A = aa_1 a_3, \quad B = a(a_3^2+a_1 a_4)+ca_1(a_3+a_5), \quad C = a_3[(a+a_1)a_4+c(a_3+a_5)];
$$
$$
a = a_1+a_2, \quad b = a_1 - a_2, \quad c = a_3 - a_5, \quad D = a_3^2 a_4, \quad a_1 = (1/2)\mu_3,
$$
$$
a_2 = (1/2)(\mu_1 \sin^2\alpha+\mu_2 \cos^2\alpha), \quad a_3 = (1/2)\mu_5, \quad \Delta = \partial_1^2 + \partial_2^2,
$$
$$
a_4 = \mu_1 \cos^2\alpha+\mu_2 \sin^2\alpha, \quad a_5 = (1/2)[\sqrt{2}(\mu_2 - \mu_1)\sin\alpha \cos\alpha + \mu_5],
$$
$$
\underset{\sim}{\mathbf{C}} = C_{IJ}\mathbf{e}_I\mathbf{e}_J, \quad \mathbf{n}^0 = n_I\mathbf{e}_I, \quad \nabla^0 = \mathbf{e}_I\partial_I, \quad \mathbf{n}^0\mathbf{n}^0\overset{2}{\otimes}\nabla^0\nabla^0 = n_I n_J \partial_I \partial_J.
$$

If we apply the operator $\underset{\sim}{\mathbf{L}}_*$ from the left with a single multiplication to the equation (the first relation in Equation (39)), and the operator $\underset{\sim}{\mathbf{T}}_*^T$ (see the corresponding relations from Equation (43)) to the boundary conditions $\underset{\sim}{\mathbf{T}} \cdot \mathbf{u} = \mathbf{P}$, then we obtain the following split equations and boundary conditions:

$$
|\underset{\sim}{\mathbf{L}}|\mathbf{u}+\underset{\sim}{\mathbf{L}}_*^T\cdot(\rho\mathbf{F})=0, \quad |\underset{\sim}{\mathbf{T}}|\mathbf{u} = \underset{\sim}{\mathbf{T}}_*^T \cdot \mathbf{P}, \tag{44}
$$

and if we look for the solution $\mathbf{u}$ in the form $\mathbf{u}=\underset{\sim}{\mathbf{L}}_*\cdot\mathbf{v}$, then we have the following split equations and boundary conditions:

$$
|\underset{\sim}{\mathbf{L}}|\mathbf{v}+\rho\mathbf{F}=0, \quad |\underset{\sim}{\mathbf{T}}|(\underset{\sim}{\mathbf{L}}_* \cdot \mathbf{v}) = \underset{\sim}{\mathbf{T}}_*^T \cdot \mathbf{P}. \tag{45}
$$

When obtaining Equations (44) and (45), we took into account the relation $\underset{\sim}{\mathbf{F}} \cdot \underset{\sim}{\mathbf{F}}_*^T = \underset{\sim}{\mathbf{F}}_*^T \cdot \underset{\sim}{\mathbf{F}} = \mathbf{E}|\underset{\sim}{\mathbf{F}}|, \; \forall \underset{\sim}{\mathbf{F}}$.

Further, applying, for example, to the split equations from (45), the *k*-th moment operator [27,32], the equations for prismatic bodies in moments with respect to any system of orthogonal polynomials can be represented in the form

$$
A\Delta^3 \overset{(k)}{\mathbf{v}} + B\Delta^2 \overset{(k)}{\mathbf{v}}'' + C\Delta \overset{(k)}{\mathbf{v}}{}^{IV} + D \overset{(k)}{\mathbf{v}}{}^{VI} + \rho \overset{(k)}{\mathbf{F}} = 0, \; k = \overline{0,\infty}, \tag{46}
$$

where in the application of the system of Legendre polynomials, the expressions for $\overset{(k)}{\mathbf{v}}''$, $\overset{(k)}{\mathbf{v}}{}^{IV}$ and $\overset{(k)}{\mathbf{v}}{}^{VI}$ are defined using the following relationship:

$$
\overset{(n)(2m)}{\mathbf{v}}(x') = (2n+1)\sum_{k=1}^{\infty} C_{k+2m-2}^{2m-1} \prod_{s=1}^{2m-1}(2n+2k+2s-1)\overset{(n+2k+2m-2)}{\mathbf{v}},
$$

$$
n \in \mathbb{N}_0, \; m \in \mathbb{N}. \tag{47}
$$

Note that analogous to Equation (46) equations with respect to **u** can be easily obtained from Equation (44). In order to shorten the letter, we will not write them out. By virtue of the first relation of (43), it is not difficult to calculate that

$$
\begin{aligned}
2I_1(\underline{\mathbf{L}}) &= (3a + b + 2a_3)\Delta + (a_4 + 2a_3)\partial_3^2, \\
2I_2(\underline{\mathbf{L}}) &= [(3a + b)a_3 + a(a + b)]\Delta^2, \quad I_3(\underline{\mathbf{L}}) = [(3a + b)a_4 \\
&\quad -2c(c + 2a_3) + (2a_3 + 3a + b)a_3]\Delta\partial_3^2 + 2(2a_4 + a_3)a_3\partial_3^4.
\end{aligned}
\tag{48}
$$

Note that the dynamic equations (equilibrium equations) in displacements for any homogeneous anisotropic body can be written in the form

$$
\underline{\underline{\mathbf{M}}}\cdot\mathbf{u} + \rho\mathbf{F} = 0 \quad (\underline{\mathbf{L}}\cdot\mathbf{u} + \rho\mathbf{F} = 0);
$$

$$
\underline{\underline{\mathbf{M}}} = \underline{\mathbf{L}} - \underline{\mathbf{E}}\rho\partial_t^2, \quad \underline{\mathbf{L}} = \sum_{k=1}^{6} \lambda_k \mathbf{w}_k\cdot\nabla\underline{\mathbf{w}}_k\cdot\nabla.
$$

It is easy to see that

$$
\begin{aligned}
I_3(\underline{\underline{\mathbf{M}}}) &= |\underline{\underline{\mathbf{M}}}| = I_3(\underline{\mathbf{L}}) - I_2(\underline{\mathbf{L}})\rho\partial_t^2 + I_1(\underline{\mathbf{L}})\rho^2\partial_t^4 - \rho^3\partial_t^6, \\
\underline{\underline{\mathbf{M}}}_* &= \underline{\underline{\mathbf{M}}}^2 - I_1(\underline{\underline{\mathbf{M}}})\underline{\underline{\mathbf{M}}} + I_2(\underline{\underline{\mathbf{M}}})\underline{\mathbf{E}} = \underline{\mathbf{L}}_* + [\underline{\mathbf{L}} - I_1(\underline{\mathbf{L}})\underline{\mathbf{E}}]\rho\partial_t^2 + \underline{\mathbf{E}}\rho^2\partial_t^4 \\
&= \underline{\mathbf{L}}^2 - \underline{\mathbf{L}}[I_1(\underline{\mathbf{L}}) - \rho\partial_t^2] + \underline{\mathbf{E}}[I_2(\underline{\mathbf{L}}) - I_1(\underline{\mathbf{L}})\rho\partial_t^2 + \rho^2\partial_t^4], \\
\underline{\mathbf{L}}_* &= \underline{\mathbf{L}}^2 - I_1(\underline{\mathbf{L}})\underline{\mathbf{L}} + I_2(\underline{\mathbf{L}})\underline{\mathbf{E}}.
\end{aligned}
\tag{49}
$$

Consequently, the decomposed equations will have the form

$$
|\underline{\underline{\mathbf{M}}}|\mathbf{u} + \underline{\underline{\mathbf{M}}}_*\cdot(\rho\mathbf{F}) = 0, \quad (|\underline{\mathbf{L}}|\mathbf{u} + \underline{\mathbf{L}}_*\cdot(\rho\mathbf{F}) = 0).
$$

Next, we represent $|\underline{\underline{\mathbf{M}}}|$ (see the first equality of Equation (49))) as a product of simple operators

$$
\begin{aligned}
|\underline{\underline{\mathbf{M}}}| &= I_3(\underline{\mathbf{L}}) - I_2(\underline{\mathbf{L}})\rho\partial_t^2 + I_1(\underline{\mathbf{L}})\rho^2\partial_t^4 - \rho^3\partial_t^6 \\
&= (b_1 - \rho\partial_t^2)(b_2 - \rho\partial_t^2)(b_3 - \rho\partial_t^2) \\
&= b_1 b_2 b_3 - (b_1 b_2 + b_1 b_3 + b_2 b_3)\rho\partial_t^2 + (b_1 + b_2 + b_3)\rho^4\partial_t^4 - \rho^3\partial_t^6; \\
I_1(\underline{\mathbf{L}}) &= b_1 + b_2 + b_3, \quad I_2(\underline{\mathbf{L}}) = b_1 b_2 + b_1 b_3 + b_2 b_3, \quad I_3(\underline{\mathbf{L}}) = b_1 b_2 b_3.
\end{aligned}
\tag{50}
$$

From this we see that $b_1$, $b_2$ and $b_3$ are eigenoperators for $\underline{\mathbf{L}}$, and $b_1 - \rho\partial_t^2$, $b_2 - \rho\partial_t^2$ and $b_3 - \rho\partial_t^2$ are the eigenoperators for $\underline{\underline{\mathbf{M}}}$.

Note that, knowing the expressions for $I_k(\underline{\mathbf{L}})$, $k = 1, 2, 3$, (the case of a transversally isotropic medium, see Equation (48)), the simple (linear) operators $b_k$, $k = 1, 2, 3$, can be found using Equation (50).

Based on the above, it is not difficult to consider other cases of anisotropic media, as well as cases of classical and micropolar isotropic materials, but in order to reduce the writing we will not dwell on them.

## 9. Quasistatic Canonical Problem of the Micropolar Theory of Elasticity in Displacements and Rotations

To reduce the letter, we consider an isotropic material with a center of symmetry. The mechanical properties of such a linear material are characterized by two fourth-rank tensors, each of which in turn is determined by three essential components. These materials are special cases of materials whose anisotropy symbols consist of three elements and the number of which, according to the classification adopted above, is 28 [29,34], among which the materials {1, 5, 3} and {5, 1, 3} are. The first of them has a positive Poisson's ratio, and the second one has negative Poisson's ratio. Under the canonical

problem we mean the problem, in the formulation of which the material tensors are written in the canonical form. We confine ourselves to the isotropic material $\{1, 5, 3\}$. In this case, the material tensors have the expressions

$$
\begin{aligned}
\underline{\underline{\mathbf{A}}} &= \frac{1}{3}(\lambda_1 - \lambda_2)\underline{\underline{\mathbf{C}}}_{(1)} + \frac{1}{2}(\lambda_2 + \lambda_7)\underline{\underline{\mathbf{C}}}_{(2)} + \frac{1}{2}(\lambda_2 - \lambda_7)\underline{\underline{\mathbf{C}}}_{(3)}, \\
\underline{\underline{\mathbf{D}}} &= \frac{1}{3}(\mu_1 - \mu_2)\underline{\underline{\mathbf{C}}}_{(1)} + \frac{1}{2}(\mu_2 + \mu_7)\underline{\underline{\mathbf{C}}}_{(2)} + \frac{1}{2}(\mu_2 - \mu_7)\underline{\underline{\mathbf{C}}}_{(3)}.
\end{aligned}
\tag{51}
$$

Then, by virtue of (51), it is not difficult to obtain equations analogous to (34), from which in the case of a quasistatic we have

$$
\Delta^2\left[\frac{1}{12}(\lambda_1 + 2\lambda_2)(\lambda_2 + \lambda_7)(\mu_2 + \mu_7)\Delta - \frac{1}{3}\lambda_2\lambda_7(\lambda_1 + 2\lambda_2)\right]\mathbf{u} + \mathbf{S}^* = 0,
$$

$$
\Delta\left\{\frac{1}{12}(\mu_1 + 2\mu_2)(\mu_2 + \mu_7)(\lambda_2 + \lambda_7)\Delta^2 - \lambda_7\left[\frac{1}{3}\lambda_2(\mu_1 + 2\mu_2)\right.\right.
$$

$$
\left.\left. + \frac{1}{2}(\mu_2 + \mu_7)(\lambda_2 + \lambda_7)\right]\Delta + 2\lambda_2\lambda_7^2\right\}\boldsymbol{\varphi} + \mathbf{H}^* = 0,
\tag{52}
$$

$$
\begin{aligned}
\mathbf{S}^* &= \lambda_7 Q_1^*(\underline{\underline{\mathbf{C}}} \cdot \nabla) \cdot (\rho\mathbf{m}) + [\mathbf{E}Q_1^* Q_4^* - (dQ_4^* - \lambda_7^2)\nabla\nabla] \cdot (\rho\mathbf{F}), \\
\mathbf{H}^* &= \lambda_7 Q_3^*(\underline{\underline{\mathbf{C}}} \cdot \nabla) \cdot (\rho\mathbf{F}) + [\mathbf{E}Q_2^* Q_3^* - (mQ_2^* - \lambda_7^2)\nabla\nabla] \cdot (\rho\mathbf{m}), \\
Q_1^* &= (b + d)\Delta, \quad Q_2^* = b\Delta, \quad Q_3^* = (g + m)\Delta - l, \quad Q_4^* = g\Delta - l, \\
d &= \frac{1}{6}(2\lambda_1 + \lambda_2) - \frac{1}{2}\lambda_7, \quad l = 2\lambda_7, \quad b = \frac{1}{2}(\lambda_2 + \lambda_7), \\
m &= \frac{1}{6}(2\mu_1 + \mu_2) - \frac{1}{2}\mu_7, \quad g = \frac{1}{2}(\mu_2 + \mu_7).
\end{aligned}
\tag{53}
$$

For $\alpha = 0$, that is, in the case of a reduced medium from Equations (52) and (53) we obtain

$$
\Delta^2\mathbf{u} + \mathbf{G} = 0, \quad \Delta^2\boldsymbol{\varphi} + \mathbf{H} = 0,
\tag{54}
$$

$$
\mathbf{G} = \frac{1}{\lambda_2(\lambda_1 + 2\lambda_2)}[2\mathbf{E}(\lambda_1 + 2\lambda_2)\Delta - (2\lambda_1 + \lambda_2)\nabla\nabla] \cdot (\rho\mathbf{F}),
$$

$$
\mathbf{H} = \frac{1}{(\mu_1 + 2\mu_2)(\mu_2 + \mu_7)}[2\mathbf{E}((\mu_1 + 2\mu_2)\Delta - (2\mu_1 + \mu_2 - 3\mu_7)\nabla\nabla] \cdot (\rho\mathbf{m}),
$$

where the first relation of Equation (54) is a classical equation, and the second has a similar form.

## 10. The Quasistatic Canonical Problem of the Micropolar Theory of Prismatic Bodies of Constant Thickness in Displacements and Rotations and in the Moments of Displacement and Rotation Vectors

Let us consider a prismatic body of constant thickness $2h$. As the base plane, we take the middle plane. Then in this case $g_M^{\hat{P}} = \delta_M^P$, $g_{\hat{P}}^3 = 0$, $g^{33} = h^{-2}$ and the nabla-operator $\hat{\nabla}\mathbb{F}$, the Laplacian $\hat{\Delta}$, $\hat{\Delta}^2$ and $\hat{\Delta}^3$ are represented in the form

$$
\begin{aligned}
\hat{\nabla}\mathbb{F} &= (\mathbf{r}^P\partial_P + \mathbf{r}^3\partial_3)\mathbb{F} = (\mathbf{r}^P\partial_P + h^{-1}\mathbf{n}\partial_3)\mathbb{F}, \quad -1 \leq x^3 \leq 1, \\
\hat{\Delta}\mathbb{F} &= (g^{PQ}\nabla_P\nabla_Q + g^{33}\partial_3^2)\mathbb{F} = (\bar{\Delta} + h^{-2}\partial_3^2)\mathbb{F}, \quad \bar{\Delta} = g^{PQ}\nabla_P\nabla_Q, \\
\hat{\Delta}^2 &= \bar{\Delta}^2 + \frac{2}{h^2}\bar{\Delta}\partial_3^2 + \frac{1}{h^4}\partial_3^4, \quad \hat{\Delta}^3 = \bar{\Delta}^3 + \frac{3}{h^2}\bar{\Delta}^2\partial_3^2 + \frac{3}{h^4}\bar{\Delta}\partial_3^4 + \frac{1}{h^6}\partial_3^6.
\end{aligned}
\tag{55}
$$

By the corresponding formula (55), Equation (52) for the theory of prismatic bodies of constant thickness in displacements and rotations can be written in the form

$$[\bar{\Delta}^3 + A\bar{\Delta}^2 + \frac{1}{h^2}(3\bar{\Delta} + 2A)\bar{\Delta}\partial_3^2 + \frac{1}{h^4}(3\bar{\Delta} + A)\partial_3^4 + \frac{1}{h^6}\partial_3^6]\hat{\mathbf{u}}$$
$$+\hat{\mathbf{S}}^{**} = 0,$$
$$[\bar{\Delta}^3 + (B\bar{\Delta}+A)\bar{\Delta} + \frac{1}{h^2}[(3\bar{\Delta}+2B)\bar{\Delta}+C]\partial_3^2 + \frac{1}{h^4}(3\bar{\Delta}+B)\partial_3^4 + \frac{1}{h^6}\partial_3^6]\hat{\boldsymbol{\varphi}}$$
$$+\hat{\mathbf{H}}^{**} = 0; \tag{56}$$

$$\hat{\mathbf{S}}^{**} = \frac{12\hat{\mathbf{S}}^*}{(\lambda_1+2\lambda_2)(\lambda_2+\lambda_7)(\mu_2+\mu_7)}, \quad \hat{\mathbf{H}}^{**} = \frac{12\hat{\mathbf{H}}^*}{(\mu_1+2\mu_2)(\mu_2+\mu_7)(\lambda_2+\lambda_7)},$$
$$A = -\frac{4\lambda_2\lambda_7}{(\lambda_2+\lambda_7)(\mu_2+\mu_7)}, \quad C = \frac{24\lambda_2\lambda_7^2}{(\mu_1+2\mu_2)(\mu_2+\mu_7)(\lambda_2+\lambda_7))},$$
$$B = -\frac{2\lambda_7[2\lambda_2(\mu_1+2\mu_2)+3(\lambda_2+\lambda_7)(\mu_2+\mu_7)]}{(\mu_1+2\mu_2)(\mu_2+\mu_7)(\lambda_2+\lambda_7)}.$$

Applying the *k*-th moment operator of some system of orthogonal polynomials (Legendre, Tchebyshev) to the equations in (56), we find the following equations for the micropolar theory of prismatic bodies of constant thickness in the moments of the displacement and rotation vectors:

$$[\bar{\Delta}^3 + A\bar{\Delta}^2]\overset{(k)}{\mathbf{u}} + \frac{1}{h^2}(3\bar{\Delta}+2A)\bar{\Delta}\overset{(k)}{\mathbf{u}}'' + \frac{1}{h^4}(3\bar{\Delta}+A)\overset{(k)}{\mathbf{u}}{}^{IV} + \frac{1}{h^6}\overset{(k)}{\mathbf{u}}{}^{VI}$$
$$+\overset{(k)}{\mathbf{S}}{}^{**} = 0,$$
$$[\bar{\Delta}^3 + (B\bar{\Delta}+A)\bar{\Delta}]\overset{(k)}{\boldsymbol{\varphi}} + \frac{1}{h^2}[(3\bar{\Delta}+2B)\bar{\Delta}+C]\overset{(k)}{\boldsymbol{\varphi}}'' + \frac{1}{h^4}(3\bar{\Delta}+B)\overset{(k)}{\boldsymbol{\varphi}}{}^{IV}$$
$$+\frac{1}{h^6}\overset{(k)}{\boldsymbol{\varphi}}{}^{VI} + \overset{(k)}{\mathbf{H}}{}^{**} = 0, \ k \in \mathbb{N}_0. \tag{57}$$

Having Equation (57), by the formula (47) it is easy to obtain systems of equations of any approximation in moments with respect to the system of Legendre polynomials. We note that the equations of the fifth (in the classical case) and the 8th (in the micropolar case) approximations in moments were obtained in the papers [27,28,32] for isotropic material in the traditional form, as well as the similarly to Equations (52), (54), (56) and (57) in the traditional form are given in [27,28,32]. Equations in moments for thin bodies with two small dimensions and thin multi-layered structures are also given there.

Adding the corresponding canonical boundary conditions to Equation (57), we obtain a canonical statement of quasistatic boundary value problems for prismatic bodies. In order to shorten the letter, we shall not dwell on this in this paper, but refer to the interested reader in the papers [27,32], in which the formulations of boundary-value problems in the traditional form are given in detail, and they easily extend to canonical statements.

It should be noted that for the theory of thin bodies, the decomposed equations at equilibrium, depending on the order of approximation, are equations of elliptical type of high order [27,28,32] and using the Vekua method [36], for them it is possible to write out analytical solutions. Note also that some questions about the application of problems on eigenvalues of tensor objects are set out in the works [37–40].

Note that using canonical representations of material objects, based on differential statements of initial-boundary value problems, it is easy to obtain variational statements, and then the application of this method can be generalized to the case of media considered, for example, in [41–44].

## 11. Conclusion

The statement of the eigenvalue problem for TBM of any order and any even rank is formulated, and also particular cases are considered. The canonical form the specific deformation energy and the CR are written. The equations of motion of a micropolar arbitrarily anisotropic medium and the boundary conditions are obtained by means of the introduced TBM operators. The formulations of initial-boundary value problems in terms of the introduced TBM operators for an arbitrary anisotropic medium are given. In particular, the initial-boundary value problems of the micropolar (classical) theory of elasticity are presented with the help of the introduced TBM operators (tensors–operators). In the case of an isotropic micropolar elastic medium (isotropic and transversely isotropic classical media) with the constructed TBM operator (tensors–operators) of cofactors to TBM operator (tensors–tensors) of the initial-boundary value problems the initial-boundary value problems are decomposed. The determinant and the cofactor tensor for the sum of six tensors are obtained. From three-dimensional decomposed initial-boundary value problems, the corresponding decomposed initial-boundary value problems for the theories of thin bodies are given.

**Author Contributions:** M.N. wrote the most part of the paper. A.U. obtained equations for the prismatic bodies.

**Acknowledgments:** This work was supported by the Russian Foundation for Basic Research, grant no. 18-29-10085-mk.

**Conflicts of Interest:** The authors declare no conflict of interest.

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
