# Peer review of "Some Applications of Eigenvalue Problems for Tensor and Tensor–Block Matrices for Mathematical Modeling of Micropolar Thin Bodies"

_mca, doi:10.3390/mca24010033_

Round 1

Reviewer 1 Report

The statement of the eigenvalue problem for a tensor-block matrix (TBM) of any order and of any even rank is formulated, and also some of its special cases are considered. The case of the motion of a particolar class of micro-polar media has been addressed. The formulations of initial-boundary value problems in terms of the introduced TBM operators for an arbitrary anisotropic medium are given. The paper is difficult to follow and it is not clear the reason why the formulation of initial-boundary value problems with this method is better than the standard variational procedures, see e.g. [A,B] for conservative and [C,D] for non conservative systems.

[A] Boutin C., dell'Isola F., Giorgio I., Placidi L (2017). Linear pantographic sheets: Asymptotic micro-macro models identification. MATHEMATICS AND MECHANICS OF COMPLEX SYSTEMS, vol. 5, p. 127-162, ISSN: 2326-7186

[B] Ugo Andreaus, Francesco dell'Isola, Ivan Giorgio, Placidi L, Lekszycki, T., Rizzi, N.L. (2016). Numerical simulations of classical problems in two-dimensional (non) linear second gradient elasticity. INTERNATIONAL JOURNAL OF ENGINEERING SCIENCE, vol. 108, p. 34-50, ISSN: 0020-7225, doi: 10.1016/j.ijengsci.2016.08.003

[C] Placidi L, Barchiesi Emilio, Misra Anil (2018). A strain gradient variational approach to damage. A comparison with damage gradient models and numerical results.. MATHEMATICS AND MECHANICS OF COMPLEX SYSTEMS, vol. 6, p. 77-100, ISSN: 2326-7186, doi: dx.doi.org/10.2140/memocs.2018.6.77

[D] Placidi L, Barchiesi Emilio (2018). Energy approach to brittle fracture in strain gradient modelling. PROCEEDINGS - ROYAL SOCIETY. MATHEMATICAL, PHYSICAL AND ENGINEERING SCIENCES, vol. 474, 20170878, ISSN: 1471-2946, doi: 10.1098/rspa.2017.0878

Author Response

The authors thank the reviewer for valuable advice that improves the work.

According to the authors, the formulation of initial-boundary problems can be done both in differential form and in variational form. From the differential formulation of the initial-boundary problems, of course, we can get the variational formulation and vice versa. We pointed about it at the end of section 1.10.  In this work, we use differential formulation. 

Reviewer 2 Report

The manuscript deals with the formulation of the eigenvalue problem for Tensor Block Matrices of any order and any even rank. In particular, after introducing the governing equations of the linear micropolar theory of elasticity, the authors discuss the decomposition of the equation of motion and boundary conditions. Finally, the quasistatic canonical problems of the micropolar theory of elasticity and of prismatic bodies of constant thickness are formulated. In general, the manuscript is presented well. However, there are some points which the authors should address:

1.       The addition of an Introduction section with a brief state of the art, objectives and novelties of the manuscript, would be appreciated.

2.       It is not clear what are the novel contributions of the present work. The majority of the results of the manuscript seem to be already obtained, as stated by the authors, in [1-5,10]. The authors should consider to highlight what are the differences with these works and highlight the novelties of the present contribution.

3.       In the introduction of the constitutive relations (2) the authors should consider to add some references, e.g. Eringen 1999.

4.       The authors should consider to specify the notation they are using, e.g. for the product $\otimes^p$, sub-tilde, sub-tilde-sub-bar, etc.

5.       Why is necessary to introduce E in equation (7)? It could be omitted and we still obtain (5). The authors should consider the discussion of the necessity of E in a more general framework.

6.       Comparing equations (10) and (8), why the \otimes operator is not reflected in (10)?

In addition, the authors may take into consideration the following minor comments:

1.       After equation (1) I think is missing U_k =

2.       Before Section 4 and at the end of the paragraph at line number 50, it should be $S_1\capS_2=\emptyset$.

3.       In equation (34), it seems that the \tilde is missing in M.

4.   At page 10 the authors state that: “… We give below the expressions for the determinant and the cofactors of the sum of six second-rank tensors without proofs”. Can the authors provide where this proof can be found?

5.       At page 10, change “tensot” for tensor.

6.       At page 12, the authors may consider to change “analogous (43) equations can” to analogous equations to (43).

7.       At page 14, after “2h” add a space.

8.       The first equation after (34) seems to be redundant.

Author Response

The authors thank the reviewer for valuable advice that improves the work and for the attentive reading of our work.

1.      The addition of an Introduction section with a brief state of the art, objectives and novelties of the manuscript, would be appreciated.

Point 1. The authors also added the section "Introduction".

2.      It is not clear what are the novel contributions of the present work. The majority of the results of the manuscript seem to be already obtained, as stated by the authors, in [1-5,10]. The authors should consider to highlight what are the differences with these works and highlight the novelties of the present contribution.

Point 2. Regarding the differences between [1-5,10] and the present work, in [1-5,10] the author presented some questions of tensor calculus. In particular, the canonical representations of the tensor and tensor-block matrix of any even rank are given, which are used in this work to model the deformation of micropolar thin bodies (to construct the canonical theory of classical and micropolar thin bodies).

3.      In the introduction of the constitutive relations (2) the authors should consider to add some references, e.g. Eringen 1999.

Point 3. According to the advice of the reviewer, the authors added a link to the monograph of Eringen when presenting the constitutive relations.

4.      The authors should consider to specify the notation they are using, e.g. for the product $\otimes^p$, sub-tilde, sub-tilde-sub-bar, etc.

Point 4. The authors left the sign of the inner p-product unchanged.

5.      Why is necessary to introduce E in equation (7)? It could be omitted and we still obtain (5). The authors should consider the discussion of the necessity of E in a more general framework.

Point 5. We introduced the unit tensor in (7) because we considered the algebraic sum of two second rank tensors. Hence, the unit tensor is necessary here.

6.      Comparing equations (10) and (8), why the \otimes operator is not reflected in (10)?

Point 6. \omits is the sign of the tensor product, and often in order to shorten the formulas, we drop this sign. However, taking into account the wishes of the reviewer, we added this sign in (8) and (10).

In addition, the authors may take into consideration the following minor comments:

The authors corrected all inaccuracies indicated below by the reviewer. 

1.       After equation (1) I think is missing U_k =

2.       Before Section 4 and at the end of the paragraph at line number 50, it should be $S_1\capS_2=\emptyset$.

3.       In equation (34), it seems that the \tilde is missing in M.

4.   At page 10 the authors state that: “… We give below the expressions for the determinant and the cofactors of the sum of six second-rank tensors without proofs”. Can the authors provide where this proof can be found?

Point 4. “… We give below the expressions for the determinant and the cofactors of the sum of six second-rank tensors without proofs”.  The proof of these formulas for two tensors is given, for example, in Pobedrya, B. E. “Lectures on tensor analysis”, 1986, (in Russian), which can be easily generalized to the algebraic sum of three tensors, and then to the sum of six tensors. The proof of these formulas is given in the work of Nikabadze M., which is accepted for publication and will be published soon in the Moscow University Bulletin.

5.       At page 10, change “tensot” for tensor.

6.       At page 12, the authors may consider to change “analogous (43) equations can” to analogous equations to (43).

7.       At page 14, after “2h” add a space.

8.       The first equation after (34) seems to be redundant.

Thanks again to the reviewer for valuable advice.
